# Primary Prostatic Stromal Sarcoma: A Case Report and Review of the Literature

**DOI:** 10.3390/medicina60121918

**Published:** 2024-11-22

**Authors:** Enes Erul, Ömer Gülpınar, Diğdem Kuru Öz, Havva Berber, Saba Kiremitci, Yüksel Ürün

**Affiliations:** 1Division of Medical Oncology, Department of Internal Medicine, Ankara University Faculty of Medicine, Ankara 06620, Türkiye; urun@ankara.edu.tr; 2Department of Urology, Ankara University Faculty of Medicine, Ankara 06620, Türkiye; ogulpinar@ankara.edu.tr; 3Department of Radiology, Ankara University Faculty of Medicine, Ankara 06620, Türkiye; dkoz@ankara.edu.tr; 4Department of Pathology, Ankara University Faculty of Medicine, Ankara 06620, Türkiye; hberber@ankara.edu.tr (H.B.); skiremitci@ankara.edu.tr (S.K.)

**Keywords:** prostatic stromal sarcoma, prostate, sarcoma

## Abstract

*Background and Objectives*: Primary prostatic stromal sarcoma is an exceptionally rare urological malignancy, constituting less than 0.1% of all prostatic cancers. It poses a significant clinical challenge due to its aggressive behavior and poor prognosis. *Materials and Methods*: We report the case of a 34-year-old male who presented with nonspecific lower urinary tract symptoms, including dysuria and increased urinary frequency. The initial diagnostic workup, including digital rectal examination and Magnetic Resonance Imaging (MRI), revealed a lobulated lesion within the prostate. A transurethral resection (TUR) was performed for diagnostic purposes, and histopathological examination revealed a “malignant mesenchymal tumor”. The patient underwent a laparoscopic radical prostatectomy. The pathology report confirmed the diagnosis of prostatic stromal sarcoma. The postoperative follow-up, including systemic CT and MRI, showed no evidence of recurrence or metastasis thus far. *Results*: Multidisciplinary management is essential for optimizing treatment outcomes in all urologic malignancies; however, it becomes particularly challenging and crucial in rare cases such as primary prostatic stromal sarcoma. In our case, the patient benefited from a coordinated approach involving urology, pathology, and oncology, underscoring the importance of collaborative care for rare and aggressive tumors like this. This case highlights the importance of early detection, complete surgical excision, and consideration of adjuvant therapies, given the aggressive nature of the disease. The role of novel therapeutic strategies, including immunotherapy and targeted therapies, is also discussed in the context of metastatic sarcomas. *Conclusions*: This case underscores the critical need for a comprehensive, multidisciplinary approach to managing primary prostatic stromal sarcoma. Ongoing research on innovative therapies offers hope for improved outcomes in metastatic stages.

## 1. Introduction

Prostatic stromal sarcoma (PSS) is an exceptionally rare urological malignancy, accounting for less than 0.1% of all prostatic cancers. It is characterized by its highly aggressive nature and poor prognosis with extremely short overall survival time [1]. PSS is a malign mesenchymal tumor unique to the prostate, such as a “stromal tumor of uncertain malignant potential” (STUMP). They both originate from the specific prostatic stroma. However, various mesenchymal tumors of muscle origin such as leiomyoma, leiomyosarcoma, and rhabdomyosarcoma, of vascular origin such as hemangioma and angiosarcoma, of fibroblastic and myofibroblastic origin such as solitary fibrous tumor and inflammatory myofibroblastic tumor, and tumors of uncertain differentiation such as synovial sarcoma may also arise in the prostate. Differential diagnosis of mesenchymal lesions in this location is troublesome because the amount of biopsy tissue usually remains limited for definite diagnosis, and pathologists are not so familiar with these lesions due to the rarity of these tumors in this location. Furthermore, clinical findings and other diagnostic procedures including initial clinical symptoms, digital rectal examination, and imaging techniques like Magnetic Resonance Imaging (MRI) and prostate-specific antigen (PSA) levels provide only general, non-specific preoperative information [2,3]. In any case, a definite diagnosis requires postoperative pathological tissue examination. In this context, while histopathological findings decide whether the tumor is malignant or benign, ancillary methods like immunohistochemistry, in general, provide information about the tumor’s origin and specific diagnosis. Due to the rarity of PSSs, knowledge regarding their prognosis and clinical course is primarily based on case reports and small case series from single-center experiences, as documented in the literature [4,5].

This case of PSS is presented due to its extreme rarity, diagnostic complexity, and potential to inform clinical management and research. With less than 0.1% of prostatic cancers being sarcomas, this case offers valuable insights to the literature in terms of differentiating PSS from malignant and benign mimickers, in terms of both histopathological and immunohistochemical findings. It highlights the challenges posed by the lack of standardized treatment guidelines and underscores the importance of multidisciplinary care in managing rare, aggressive tumors. Additionally, this case emphasizes the need for further exploration of novel therapies, such as immunotherapy and targeted treatments, which may improve outcomes in similar rare sarcomas. This case contributes to the limited literature and may guide future treatment protocols.

## 2. Case Report

A 34-year-old male patient with no prior medical history presented to the urology clinic in March 2023 with lower urinary tract symptoms, including dysuria, increased urinary frequency, and urgency. Microscopic hematuria was detected on urinalysis, and the patient’s prostate-specific antigen (PSA) level was 4.81 ng/mL. A digital rectal examination revealed an enlarged, firm, and palpable prostate nodule without tenderness. A multiparametric prostate MRI revealed a lobulated, well-circumscribed lesion approximately 3.2 × 3 × 2.5 cm in size located in the left lobe and extending to the prostatic urethra. In T2-weighted images, the lesion displayed heterogeneous hyperintense signal characteristics. The lesion was confined to the prostate gland. No pathologically enlarged or suspicious lymph nodes were identified in the pelvic region (Figure 1).

A prostate biopsy was not performed due to the specific MRI findings and the need for a larger tissue sample to achieve an accurate diagnosis; therefore, transurethral resection of the prostate (TURP) was chosen over a standard biopsy. Pathologic examination of the TUR material revealed polypoid projections of an atypical spindle cell proliferation into the lumina of the prostatic urethra. Atypical spindle cells were invading prostatic stroma between benign-looking distorted and dilated prostatic glands, which showed proliferative and metaplastic changes. Atypical stromal tumor cells exhibited discrete histologic patterns including hypercellular, storiform, myxoid, and phyllodes-like areas. Slightly increased mitotic activity and a single atypical mitosis were reported. No malignant epithelial component (carcinoma component) was observed in the current TUR samples. Some of the samples were necrotic with hemorrhage and fibrin plugs. Immunohistochemical analysis of the tumor revealed variable positivity for Vimentin, p53, ER, and PR in the limited tumor tissue of the TUR material. Rare tumor cells showed Desmin and SMA staining. Tumor cells were negative with the wide-spectrum cytokeratin panel (HMW-CK, LMW-CK, CK8/18, CK7, CK20), and also negative for CD34, Myogenin, S100, CD117 (KIT), CD31, ALK1, and CD99. The Ki-67 proliferation index was approximately 40% in hot-spot areas. The histopathological diagnosis was a “spindle cell malignant mesenchymal tumor (sarcoma)” with a comment. Though tumors of muscle and vascular origin were excluded by the immunohistochemical panel, the findings did not allow a specific diagnosis. Differential diagnosis focused on the differentiation between prostatic stromal sarcoma and carcinosarcoma. It was important to examine the entire tumor for a carcinoma component to reliably exclude carcinosarcoma.

The patient subsequently underwent a laparoscopic radical prostatectomy. The tumor was restricted in the prostate gland and showed well-defined borders. Sampling of the entire tumor revealed additional findings for specific diagnosis. Histopathological examination showed a similar variety of proliferation patterns with additional diffuse stromal growth patterns, some with rare, bizarre, atypical tumor cells and barely noticeable atypical mitosis. The glandular epithelial component showed crowding and complexity with extensive squamous, urothelial, and intestinal metaplastic changes, but no cancerous differentiation. At high-power fields (×40 magnification), mitotic activity was slightly increased with the highest frequency of two mitoses in one magnification area. Neither tumor necrosis nor vascular invasion was noted. Additionally, immunohistochemical examination revealed focal CD34 positivity, which was negative in the previous tumor samples of TUR material. These findings were consistent with prostatic stromal sarcoma, most likely a low-grade rather than a high-grade sarcoma, based on the radical prostatectomy specimen (Figure 2 and Figure 3).

Postoperatively, the patient opted for follow-up without additional radiotherapy and chemotherapy treatment. To date, follow-up imaging, including systemic CT scans and prostate MRI at 3 month intervals, has shown no evidence of recurrence or metastasis.

## 3. Discussion

Our case represents a rare instance of primary prostatic stromal sarcoma, a malignancy that constitutes less than 0.1% of all malignant prostate tumors. The patient in our case was notably free from common risk factors for prostatic tumors such as prostatitis, perineal trauma, prior prostate biopsy, and radiation exposure, highlighting the sporadic and unpredictable nature of this malignancy [6,7]. Our patient, a 34-year-old male, falls within the typical age range for prostate sarcoma presentation, which is younger than that seen in prostate adenocarcinoma. This aligns with the existing literature, which reports most patients presenting between 30 and 40 years of age. Unfortunately, it carries a very poor prognosis, with over half of the patients succumbing to the disease within 24 months of diagnosis. In our case, no evidence of metastasis has been detected during postoperative follow-up. This is a notable finding as early metastasis is frequently observed in similar cases, most commonly involving the lungs. The absence of metastasis thus far offers hope for a more favorable outcome, although vigilant long-term monitoring remains essential [7,8]. Among the histological subtypes of sarcomas involving the prostate, leiomyosarcoma is the most frequently observed, whereas prostatic stromal sarcoma is very rare. The low-grade nature of the stromal sarcoma in our patient further differentiates this case, adding valuable insight to the limited data available on this rare malignancy [5,7].

### 3.1. Diagnostic Workup

The presentation of prostate sarcoma is nonspecific, with patients typically seeking medical attention for symptoms such as frequent urination, urgency, incomplete bladder emptying, dysuria, and either microscopic or gross hematuria. Additionally, patients may report discomfort in the perineal area and even experience constipation [9]. Digital rectal examination is not specific and is subject to the examiner’s subjective interpretation. It may reveal a firm, soft, or elastic enlarged prostate or nodule, depending on the individual case. There is no standardized guideline or definition for MRI imaging specific to prostate sarcoma, and the presence of a tumoral lesion is often inferred based on abnormal signal characteristics. Adenocarcinoma, the most common malignancy of the prostate, typically appears as a lesion with isointense in T1-weighted images and hypointense in T2-weighted images. These lesions often exhibit diffusion restriction and demonstrate early enhancement in the arterial phase. Prostatic sarcomas, however, often present with a distinct appearance on T2-weighted images, significantly different from that of typical adenocarcinomas. In MRI, primary prostatic stromal sarcoma can have characteristics that differentiate it from benign tumors. These may include central necrosis, a T2 hypointense pseudocapsule, and a fibrotic ring resulting from inflammation around the pseudocapsule. MR spectroscopic imaging can help differentiate primary prostatic stromal sarcoma from benign lesions like benign prostatic hyperplasia by demonstrating a markedly elevated choline-to-citrate ratio [10]. In our case, MR spectroscopy was not performed, but T2-weighted imaging revealed a well-defined heterogeneous mass with moderate diffusion restriction. Definitive diagnosis relies on histopathological and immunohistochemical examinations of tumor tissue.

### 3.2. Histopathological Findings

In the context of mesenchymal tumors, histopathological findings are critical when deciding whether the tumor is benign or malignant. Severe nuclear atypia, a high mitotic rate, atypical mitoses, tumor necrosis, lymphovascular invasion, and adjacent organ invasion emerge as the most noteworthy parameters indicating a malignant nature; however, histopathological patterns and cytological morphology may not always reveal distinctive features to differentiate the tumor origin. A smooth muscle-derived tumor (leomyosarcoma) may not be differentiated from a striated muscle-derived tumor (rhabdomyosarcoma), a malignant peripheral nerve sheet tumor, or a vascular tumor (angiosarcoma). Ancillary methods, including immunohistochemistry, generally provide critical information for determining the cell origin. In our case, cellular atypia, increased mitotic activity, and atypical mitoses were the main histopathological parameters to decide the malignant nature of the tumor. Vimentin positivity and the negativity of the cytokeratin panel supported the spindle cell morphology of the tumor as a mesenchymal tumor (sarcoma). Literature data show that the immunoprofile of PSS is consistently positive for Vimentin but variably positive for CD34. Besides this, variable positivity for SMA, Desmin, ER, PR, and even epithelial markers such as pancytokeratin is being reported in case series, which are already rare. In our case, CD34 positivity was found focally by the examination of the entire tumor in a radical prostatectomy specimen. Nondiagnostic staining patterns of Desmin, SMA, and Myogenin provided the exclusion of muscle-originated tumors. Eventually, appropriate histopathological findings with Vimentin and CD34 positivity made the specific diagnosis of prostatic stromal sarcoma possible (Table 1) [2]. In the literature, case reports with CD34-negative findings have also been documented; thus, while immunohistochemistry is essential for diagnosing prostatic stromal sarcoma, it should not be the sole diagnostic criterion [2]. Furthermore, in most cases, the Ki-67 mitotic index has been found to be significantly elevated, and in our case, it was 40%.

### 3.3. Treatment

The rarity of primary prostate sarcoma poses a significant challenge to conducting clinical studies, resulting in a limited amount of current data to guide treatment management [5]. As with other urologic tumors, treatment decisions for primary prostate sarcomas should be made within the framework of multidisciplinary tumor boards, involving collaboration among oncology, pathology, urology, and radiation oncology specialists. This approach ensures a comprehensive and coordinated management strategy. Although primary prostate sarcomas are rare and there are no standardized treatment protocols, complete surgical excision, when feasible, is considered the most effective treatment approach [10]. Due to the aggressive nature of the disease, radical cystoprostatectomy or prostatectomy combined with lymph node dissection should be considered as part of the surgical treatment approach. In the United States, for locally invasive prostate sarcomas, preoperative neoadjuvant chemotherapy and radiotherapy are often administered to reduce tumor size and achieve clear surgical margins, facilitating complete tumor resection. On the other hand, in China, unlike the approach in the United States, upfront surgery is often preferred, as seen in our case [4]. Additionally, patients may opt not to undergo chemotherapy and radiotherapy for various reasons.

The risk of relapse is primarily influenced by factors such as histological type, tumor grade, size, and the extent of surgical resection [5]. The primary goal in surgery should be to achieve a safe oncologic margin through complete en bloc resection. In the literature, Salih et al. reported a case report whereby, in their pathological specimen, the resected prostatic sarcoma encircled the prostatic urethra and was very close to the excision margin. They considered the positive surgical margin to be a significant risk factor for early recurrence, with a relapse observed as early as six months following adjuvant chemotherapy [18].

Reese and colleagues described a case where neoadjuvant treatment with ifosfamide and adriamycin, alongside radiotherapy, led to a complete pathological response during subsequent surgery. Despite this outcome, the risk of recurrence over time remains, and the long-term prognosis is still uncertain [15].

There is no consensus on the management of primary prostate sarcoma with adjuvant chemotherapy; however, due to the aggressive nature of the disease, it is recommended that adjuvant therapy be initiated as early as possible. Ueda et al. administered adjuvant chemotherapy with ifosfamide and doxorubicin in a case of primary stromal sarcoma. In the follow-up period of up to 10 months post-surgery, no recurrence or metastasis was observed [8]. In another PSS case reported by Salih et al., the same chemotherapy regimen could only be administered for two cycles due to adjuvant tolerance issues, yet early recurrence was detected [18]. The potential role of neoadjuvant or adjuvant radiotherapy in PSS could be extrapolated from high-risk sarcomas with characteristics, such as high tumor volume, high grade, and the absence of R0 resection specifically for adjuvant setting. However, due to the extreme rarity of PSS, there is insufficient evidence in the literature to support these approaches definitively. Additionally, the potential for unexpected excessive bowel toxicity following radiotherapy in the postoperative setting should be carefully considered [18,21] (Table 2). Unlike prostate adenocarcinoma, there is no biological marker like PSA for diagnosis, monitoring, prognosis, or predicting recurrence in prostatic stromal sarcoma. Unfortunately, there are no established guidelines on how to monitor prostatic stromal sarcomas. However, as with other sarcomas, adult patients treated with chemotherapy and/or radiotherapy require careful long-term monitoring due to potential toxicity. Regimens containing ifosfamide and doxorubicin can lead to cardiomyopathy, renal tubular and/or glomerular dysfunction, and infertility; thus, these risks should be considered and appropriate precautions taken. The NCCN guidelines for sarcomas recommend a physical examination and medical history every three months. Due to the aggressive nature of the disease, we planned systematic imaging with CT and prostate MRI in our case to monitor for recurrence [22,23].

The 5-year survival rate is 44%, and although the disease is known to have an aggressive course and poor prognosis, the outlook can vary depending on the presence of metastasis and the status of surgical margins [19]. In case reports documented in the literature, survival ranges from as short as 6 months to as long as 8 years [1,6].

## 4. Future Directions and Novel Therapies

In recent times, new treatment options such as immunotherapy and targeted therapies have revolutionized oncology, becoming the fourth and fifth pillars alongside chemotherapy, radiotherapy, and surgery. With a deeper understanding of sarcoma subtypes and cancer biology in recent times, approaches involving immunotherapy and targeted therapies are gaining increased emphasis and significance [26].

Comprehensive studies beyond case reports do not exist in the literature, and unfortunately conducting randomized controlled trials is very challenging due to the rarity of PSS. In a study involving patients with primary prostatic stromal sarcoma, DNA and RNA sequencing revealed that these tumors are molecularly heterogeneous and might not represent a distinct biological entity apart from soft tissue and uterus. This finding suggests that the diagnostic terminology may need to be re-evaluated. More importantly, the study demonstrated that based on underlying genetic alterations, a subset of cases could possess potentially actionable molecular findings, offering new avenues for targeted therapy [27].

As a targeted therapy, the multikinase inhibitor Regorafenib has demonstrated significant antitumor activity by prolonging progression-free survival (PFS) in a double-blind, randomized Phase 2 trial that included patients with soft tissue sarcomas. Further investigation is warranted to establish its therapeutic role within the expanding range of treatment options available for these sarcomas [28].

In patients with metastatic sarcoma, the anti-PD-1 antibody nivolumab has contributed to an improvement in progression-free survival (PFS) [29]. Pembrolizumab has also been approved by the FDA for the treatment of advanced cancers with high microsatellite instability (MSI) or defects in mismatch repair (dMMR). Additionally, it has recently received FDA approval for treating advanced cancers with a high tumor mutational burden [TMB-H; ≥10 mutations/megabase (mut/Mb)] regardless of tumor type. Consequently, a small subset of sarcomas exhibiting MSI, dMMR, or high TMB should be considered for immunotherapy as a tumor-agnostic approach, either as part of standard care or within clinical trials [30,31,32].

A high expression of fibroblast activation protein (FAP) is observed in several malignant solid tumors. FAPI-46, a small theranostic ligand, is used for FAP-targeted PET imaging and subsequent radioligand therapy (RLT). In a study, treatment with 90Y-labeled FAPI-46 (90Y-FAPI-46 RLT) resulted in stable disease, as observed through CT/MRI, in approximately one-third of patients with previously progressive metastatic sarcoma, pancreatic cancer, and other malignancies. Partial responses were specifically noted in sarcoma patients. This radionuclide therapy with 90Y-FAPI-46 proved to be safe and achieved tumor control in a subset of patients. Further prospective studies are needed to refine and evaluate RLT in patients with metastatic sarcoma [33].

Research suggests that in primary prostate sarcoma, the Prostate-Specific Membrane Antigen (PSMA) may also be secreted, presenting a potential therapeutic target. PSMA PET has become established in clinical practice for patients with biochemical recurrence after primary treatment and staging high-risk prostate cancers. Moreover, PSMA-targeted ligands labeled with therapeutic radionuclides like lutetium-177 or actinium-225 are being used in advanced prostate adenocarcinoma. Despite its name, the Prostate-Specific Membrane Antigen (PSMA) is not exclusive to the prostate; it is a type 2 transmembrane glycoprotein. High PSMA expression has been observed in aggressive and advanced sarcomas in cases in the literature, suggesting the potential for therapeutic targeting with radioligands [34]. Chen et al. reported a case of primary prostate sarcoma in which 18F-prostate-specific membrane antigen (PSMA) PET/CT scans at the time of diagnosis revealed high PSMA expression in both the prostate mass and multiple liver lesions. After eight cycles of etoposide and VDC (vincristine, dactinomycin, and cyclophosphamide), follow-up PSMA PET scans showed a reduction in uptake, indicating a positive response to chemotherapy [24]. However, these observations are primarily case-based, and broader, future studies are needed to validate these findings [35].

Furthermore, defects in homologous recombination repair (HRR) could serve as a basis for innovative treatment approaches in primary prostate sarcomas [26]. Particularly in ovarian, breast, prostate adenocarcinomas, and pancreatic cancers, homologous recombination repair defects, such as BRCA1-2 loss or mutation, can be observed, making the tumors sensitive to treatment with oral small-molecule poly-ADP-ribose polymerase (PARP) inhibitors that induce double-strand breaks. Recently, uterine leiomyosarcomas, a sarcoma subtype, have emerged as candidates for characterization by homologous recombination repair defects and BRCA-2 loss. Ongoing studies are exploring the potential to identify DNA repair defects in other sarcoma types as well, offering the hope of targeted therapy opportunities [26].

## 5. Conclusions

In summary, the insights gained from this case of primary prostatic stromal sarcoma emphasize the importance of a multidisciplinary treatment approach, the potential sufficiency of surgical management in low-grade cases, and the promise of novel therapies in rare sarcoma management. These lessons could inform future clinical decision-making and contribute to the development of more refined treatment protocols for rare sarcomas.

## Figures and Tables

**Figure 1 medicina-60-01918-f001:**
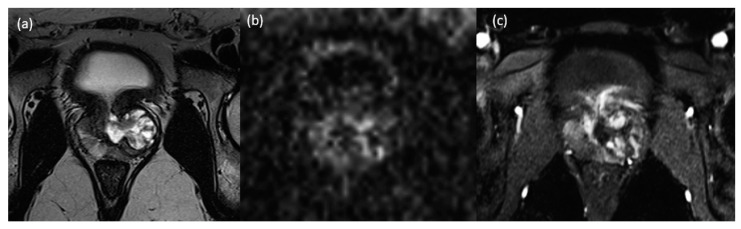
(**a**–**c**). In the T2-weighted image (**a**), a heterogeneous, well-defined, lobulated nodular lesion is observed, confined to the prostate gland, exhibiting marked hyperintense signal characteristics originating from the midline urethral area and largely filling the left lobe of the prostate gland. In the diffusion-weighted image (**b**) acquired with a b-value of 2000, focal areas of diffusion restriction are observed within the lesion. The lesion demonstrates intense and heterogeneous enhancement in the early arterial phase (**c**).

**Figure 2 medicina-60-01918-f002:**
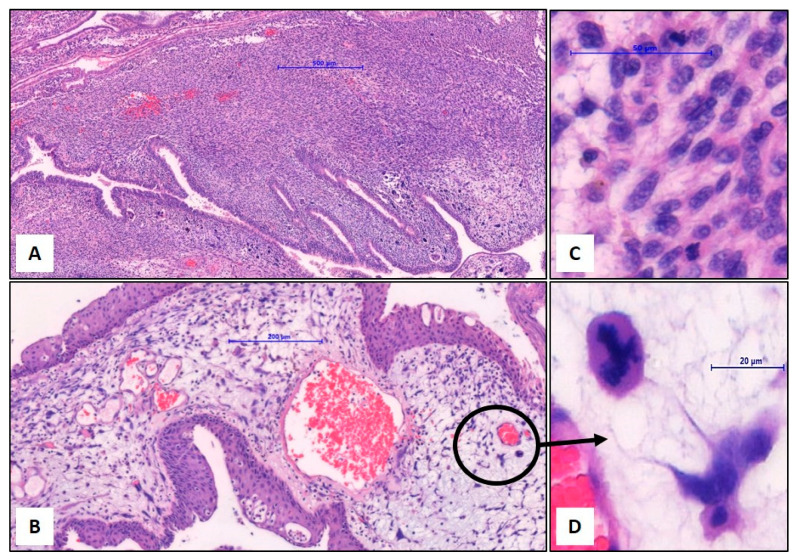
Histopathological findings of prostatic stromal sarcoma. (**A**) Invasion of the atypical spindle tumor cells between the metaplastic benign epithelial and glandular structures in a diffuse stromal growth pattern with phyllodes-like and hypercellular areas, ×40, Hematoxylin–Eosin. (**B**) Hypocellular myxoid areas with round and starry tumor cells, ×100. (**C**) Increased mitotic activity in high magnification, ×400. (**D**) Remarkable atypical tripolar mitosis, ×690.

**Figure 3 medicina-60-01918-f003:**
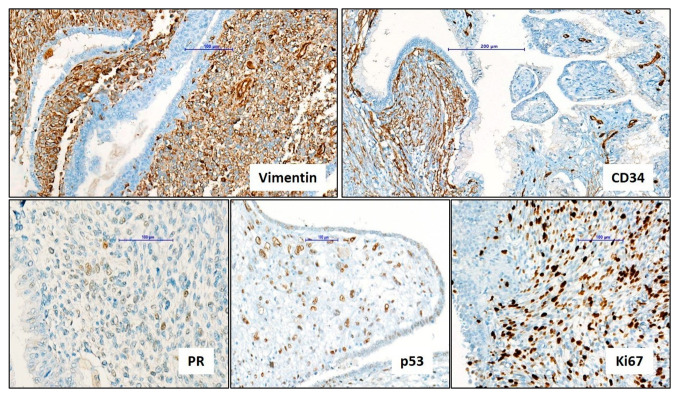
Immunohistochemical findings of prostatic stromal sarcoma. Diffuse and strong positivity with Vimentin (×150); focal positivity with CD34 (vascular structures as internal control) (×110); infrequent positivity with PR (×220); diffuse positivity with p53 (×150); high proliferation index with Ki67 (×150).

**Table 1 medicina-60-01918-t001:** Summary of patient’s pathological results in the literature.

Cases	Surgical Margin	Pathological Diagnosis	Immunohistochemistry
Our case	Microscopically negative	Prostatic stromal sarcoma	Vimentin (+); CD34, ER and PR (focally,+), Desmin and SMA (rare cell +); p53(+); CK panel, EMA, S100, CD99, Myogenin, CK19, BCL-2 (−); Ki67 index 40%
Ueda et al. [8]	Microscopically negative	Prostatic stromal sarcoma	Vimentin, CD34(+); KI67 index 20%, PSA, C-KIT, stat 6 (−)
Yang et al. [11]	Microscopically negative	Prostatic stromal sarcoma	Vimentin(+),PSA; CD34(−);
Yirui Wei et al. [1]	Microscopically negative	High-grade sarcoma	Vimentin(+)KI67 (index approximately 90%); CK; Desmin (focally +); CK7; P63; EMA; SMA; CD34; BCL2; CD99; S100; CD31; MYOD1(−)
Yirui Wei et al. [1]	Microscopically negative	High-grade sarcoma/stromal sarcoma	Vimentin(+); CD34; CD31; BCL-2 (partially+); FVIII; SMA; Desmin; S-100; EMA; CK; CK19; CK7; CD99; ER(−)
Yirui Wei et al. [1]	Microscopically negative	Spindle cell sarcoma	Vimentin(+); BCL-2(+); KI67(index approximately 40%); Desmin (partially +); CK; S100; CD99; EMA; Myogenin; PgR; ER; CK19; SMA; CD34(−)
Froehner et al. [12]	Microscopically negative	High-grade stroma sarcoma	BCL-2, CD99, CD117 (+), CD34(−); CK-pan (−)
Xu et al. [13]	Microscopically negative	Spindle cell undifferentiated sarcoma	Vimentin(+), TLE (+), epithelial membrane antigen, AE1/3, Cam5.2, SATB2, and CD34 (focally +). SOX10, S100 protein, P63, cytokeratin5/6, NKX3.1, PSA, alpha-methylacyl-CoA racemase (AMACR), PgR, STAT6, CD117, DOG1, cyclinD1, smooth muscle actin, desmin, myogenin, HMB45, and claudin-4 (−) KI67 index approximately 50%, ARID1A deficient (IHC and NGS confirmed)
Ruihe et al. [14]	Microscopically negative	Prostatic stromal sarcoma	CD34, S-100 (+),Ki-67 (hot spot 40%+), SMA, Desmin, Myogenin, Myo D1,CK-pan, P504s, and SATB2 (−).
Ronchi et al. [6]	Microscopically negative	Malignant solitary fibrous tumor of the prostate	CD34, BCL-2, CD99, and STAT6 (+), PgR partially positive, CK, SMA, desmin, S100, EMA, calponin, CD117, ER, and B-catenin (−)
Reese et al. [15]	The surgical margins were positive after the initial subtotal prostatectomy, but no residual sarcoma was found in the cystoprostatectomy specimen following neoadjuvant therapy.	High-grade stromal sarcoma	CD117 and MDM2(+)
Öztürk et al. [7]	NA	Spindle cell sarcoma	Vimentin, SMA, and desmin(+), CD34, S100, and progesterone receptors(−). The Ki-67 proliferation index was 2%.
Huh et al. [9]	NA	primary prostatic extra-GIST	KIT (CD117) and CD34(+),desmin, SMA, CK, and S-100(−)
Zhang et al. [16]	NA	primary prostatic extra-GIST	DOG-1, CD117 (c-KIT), and CD34(+), S-100, SMA(−) The mitotic count was more than 10 per 50 HPF.
Zampareseet al. [17]	Microscopically negative	Low-grade sarcoma/stromal sarcoma	CD34 (+) and PgR (focal +), CK, desmin, S-100, Bcl-2, chromogranin, CD117, and SMA(−) Very low expression of p53 and Ki-67.
Salih FM et al. [18]	Very close to its excision margin	High-grade stromal sarcoma	CD117, CD34, ER, and PgR (-)Ki-67 proliferation index was 25%.
Rao et al. [19]	NA	High-grade stromal sarcoma	Vimentin, CD34, PgR, and SMA (focal +) Pancytokeratin, AMACR, h-caldesmon, myogenin, DOG-1, CD117 (-)
Ohashi et al. [20]	NA	Prostatic stromal sarcoma	Vimentin, CD34 (+)PgR (partly +) Desmin, Myo-D1, SS18-SSX, AE1/3, PSA, S100, and SMA(−)

Abbreviations: BCL, B-cell lymphoma; CD, cluster of differentiation; CK, cytokeratin; EMA, epithelia membrane antigen; ER, estrogen receptor; MYOD, myoblast determination protein; SMA, smooth muscle actin; PgR, progesterone receptor; NA, non-available; GIST, gastrointestinal stromal tumor; HPF, high-power field.

**Table 2 medicina-60-01918-t002:** Clinical information of patients with primary prostatic stromal sarcoma in the literature.

Cases	Age	Symptom at Diagnosis	AJCC Stage	PSA Level at Diagnosis (ng/mL)	Surgical Procedure	Treatment	Metastatic Status at İnitial Presentation	Final Status	Survival Time
Our case	34	Dysuria, Microscopic hematuria	III	4.81	LRP	None	None	No evidence of recurrence	15 months
Yirui Wei et al. [1]	27	Microscopic hematuria	IV	2.23	LRP	NA	Bilateral inguinal lymph nodes	No evidence of recurrence, lost to follow-up	6 months
Yirui Wei et al. [1]	64	Dysuria	III	0.88	LRP	None	None	Died of disease	<12 months
Yirui Wei et al. [1]	38	Dysuria	III	4.86	LRP	None	None	No evidence of recurrence, lost to follow-up	6 months
Ueda et al. [8]	40	Urinary distention	III	0.62	Total pelvic exenteration	Adjuvant 4 cycle doxorubicin and ifosfamide	None	No evidence of recurrence	10 months
Yang et al. [11]	49	Dysuria, Microscopic hematuria	III	0.64	RP	None	None	NA	NA
Chen et al. [24]	23	Dysuria	IV	1.08	None	8 cycle etoposide and VDC (vincristine, dactinomycin, cyclophosphamide) + external radiation therapy	Multiple liver metastasis	NA	NA
Yang et al. [25]	58	Difficulty in urination	III	0.77	RP	None	None	No evidence of recurrence	12 months
Xu et al. [13]	58	Dysuria, Difficulty in urination	III	3	LRP with bilateral pelvic lymphadenectomy	Etoposide and cisplatin for 2 months before surgery doxorubicin and ifosfamide for 5 months	None	Bilateral inguinal lymph nodes, bilateral obturator muscles, sigmoid colon, and rectum metastasis	9 months
Froehner et al. [12]	31	Gross hematuria	III	NA	Pelvic exenteration followed by sigmoidorectostomy and ileum neobladder construction	None	None	widespread metastases occurred 4 months after surgery	NA
Ruihe et al. [14]	17	Hematuria, abdominal pain	IV	NA	NA	NA	Multiple pulmonary nodules, bone lesions, pelvic lymph nodes	NA	NA
Ronchi et al. [6]	62	urinary retention, constipation	III	5.8	RP	None	None	No evidence of recurrence	8 years
Hicks et al. [10]	66	urinary retention	III	NA	TURP followed by an open radical cystoprostatectomy with retroperitoneal lymph node dissection and urinary diversion	None	None	NA	NA
Reese et al. [15]	66	Obstructive lower urinary tract symptoms	III	3.5	Robotic-assisted suprapubic prostatectomy, followed by radical cystoprostatectomy with intraoperative radiation therapy	Neoadjuvant chemotherapy (ifosfamide and adriamycin) and radiation (50 Gy of intensity-modulated radiation therapy), followed by radical cystoprostatectomy with additional intraoperative radiation therapy (10 Gy)	No evidence of distant metastasis; however, a peritoneal tumor implant was later identified	Pathology from the cystoprostatectomy specimen showed no residual sarcoma; however, an 8 cm high-grade pleomorphic sarcoma consistent with the primary tumor was found in the resected peritoneal mass.	NA
Öztürk et al. [7]	39	Obstructive lower urinary tract symptoms	IV	0.5	NA	Doxorubicin-based chemotherapy	Widespread lung and liver metastases	NA	NA
Huh et al. [9]	50	Weak urinary stream, sensation of residual urine, and perineal discomfort	III	0.85	NA	Planned radical prostatectomy (patient refused surgery); no neoadjuvant treatment with imatinib due to cost and lack of insurance	None	no follow-up was conducted	Not specified due to lack of follow-up
Zhang et al. [16]	31	Dysuria, urinary frequency and urgency, and intermittent gross hematuria	III	0.37	No surgical treatment was performed due to the tumor’s extension and risk of rectal injury.	Administered imatinib (400 mg per day) intermittently due to financial reasons, resulting in poor response.	None	The patient developed intestinal obstruction and died from electrolyte disturbances and multiple organ failure.	The patient survived for a few months after diagnosis, but no exact survival time is provided.
Zamparese et al. [17]	71	Urinary obstruction symptoms	III	NA	TURP	None	None	No evidence of recurrence	15 months
Salih FM et al. [18]	21	urinary retention	III	2	Radical cystoprostatectomy	Adjuvant 2 cycle doxorubicin and ifosfamide	None	A pelvic mass recurrence causing right-sided hydronephrosis was observed six months after adjuvant chemotherapy.	17 months
Rao et al. [19]	50	dysuria	IV	NA	Radical cystoprostatectomy	None	İliac and inguinal lymph nodes	NA	NA
Ohashi et al. [20]	23	Gross hematuria, hematospermia	III	1.12	Total pelvic exenteration and construction of ileal conduit and colostomy	None	None	Lung and pelvic lymph node metastasis was observed two months after surgery.	NA

Abbreviations: AJCC, The American Joint Committee on Cancer; LRP, laparoscopic radical prostatectomy; PSA, prostate-specific antigen; NA, non-available; TURP, Transurethral resection of the prostate.

## Data Availability

Data are contained within the article.

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
