# Peer review of "Primary Prostatic Stromal Sarcoma: A Case Report and Review of the Literature"

_medicina, 2024, doi:10.3390/medicina60121918_

Round 1

Reviewer 1 Report

Comments and Suggestions for Authors

This case report offers valuable insights into the diagnosis, management, and future therapeutic directions for primary prostatic stromal sarcoma (PSS), a highly rare and aggressive malignancy of the prostate that accounts for less than 0.1% of all prostatic cancers. The report documents the clinical presentation, histopathology, immunohistochemistry, treatment, and follow-up for a 34-year-old male patient, adding depth to the limited literature on PSS. Here are a few comments meant to improve the quality of the paper:

1. Literature Review and Contextualization

While the paper mentions the rarity of PSS, it could better contextualize this case by providing a more robust review of existing cases and recent advancements. Although some comparative references are provided, these are limited, and there is insufficient synthesis of previous findings on clinical outcomes, immunohistochemical profiles, and survival rates. Additionally, references to some emerging treatments, like immunotherapy, are cited but not well-supported with relevant evidence or trials.

Improvement: Expanding the literature review to include recent case studies, clinical trials, and larger sarcoma studies would add depth. Including a comparative table summarizing existing cases would also enhance clarity and relevance.

2. Diagnostic Approach

The diagnostic process relies heavily on traditional methods (imaging, immunohistochemistry, histopathology), but the rationale for some diagnostic decisions is not fully explained. For example, the decision to avoid MR spectroscopy is mentioned but not justified; this technique might have provided additional differentiating characteristics.

Improvement: Detailing the rationale behind excluding or including certain diagnostic tests, such as MR spectroscopy, would strengthen the case presentation. Additionally, adding a detailed description of the histopathological characteristics observed in relation to existing diagnostic criteria for PSS would improve the diagnostic narrative.

3. Methodological Details

The paper lacks a clear explanation of the methodology used in immunohistochemical testing. Details about controls, specific markers chosen, and interpretation of staining results are missing, leaving questions about reliability and reproducibility. For instance, CD34 and Desmin positivity is mentioned, but without a methodological context or explanation of its diagnostic implications.

Improvement: Providing a more detailed immunohistochemistry protocol, including marker selection rationale and result interpretation standards, would increase the rigor and reproducibility of the findings.

4. Case Comparisons and Novelty

Although the paper mentions that this case is notable due to the patient’s younger age, it does not elaborate on how this compares with the age distribution or typical outcomes of PSS. Additionally, novelty claims regarding immunotherapy and targeted therapy discussions lack adequate evidence, especially as there is limited or no data on the efficacy of these treatments for PSS specifically.

Improvement: Strengthen the novelty of the case by clearly contrasting it with established PSS cases and adding discussions on how young patient age impacts prognosis. When discussing novel treatments, include supporting data from similar cancers or a research-based rationale for potential applicability in PSS.

5. Future Directions and Therapeutic Innovations

The paper addresses future treatment directions like immunotherapy, targeted therapies, and novel diagnostic approaches, but these discussions lack focus and specific clinical applicability. The introduction of treatments like PSMA-targeted therapy and FAPI-46 is intriguing but speculative without data on PSS-specific outcomes.

Improvement: A more targeted discussion that either emphasizes specific therapies relevant to the case or hypothesizes why certain therapies (e.g., PARP inhibitors, anti-PD-1 antibodies) might be applicable based on the tumor’s genetic profile would improve cohesion. Where applicable, the authors could advocate for clinical trials or studies to validate these emerging therapies in PSS.

6. Patient Follow-Up and Outcome Analysis

Although the patient’s postoperative course is mentioned, the discussion does not provide insight into long-term follow-up plans, such as surveillance intervals, imaging frequency, or potential markers for recurrence.

Improvement: Including a specific follow-up protocol would add practical value to the case report. Furthermore, comparing the chosen follow-up strategy with others used in similar sarcoma cases would enhance the clinical applicability of the study.

7. General Organization and Clarity

The paper’s structure could benefit from clearer section headings, as the flow between case description, histopathology, treatment approach, and future therapeutic discussion can be confusing. Additionally, the discussion contains some redundancies, and references are sparse in certain sections where data-driven claims are made.

Improvement: Reorganize the discussion to avoid overlap and create distinct subheadings for diagnostic workup, treatment decisions, histopathology findings, and future directions. Adding references, particularly in the immunohistochemical findings and therapeutic innovations sections, would substantiate claims.

Summary of Suggestions

The study could be significantly strengthened by:

A more comprehensive literature review, contextualizing the case in the broader context of PSS management.

Detailed methodological insights into immunohistochemical testing and diagnostic approaches.

Explicit descriptions of multidisciplinary involvement and follow-up plans.

A more focused discussion of future therapeutic directions with supporting data where possible.

Conclusion

This case report is a valuable addition to the sparse literature on PSS, offering insights into diagnostic approaches, the potential sufficiency of surgery in low-grade cases, and emerging therapies. It underscores the critical need for a multidisciplinary approach and highlights the potential of targeted and immunotherapies for aggressive sarcomas. However, given the rarity of PSS, continued research and documentation are essential to develop evidence-based treatment guidelines and improve patient outcomes in this challenging malignancy.

Author Response

We want to thank the reviewers for their constructive criticism and insightful comments. We diligently worked to address each of these comments constructively. This manuscript has been read and approved by all the authors. Our responses to the reviewer’s comments are given point by point:

Reviewer(s)' Comments to Author:

Reviewer 1:

This case report offers valuable insights into the diagnosis, management, and future therapeutic directions for primary prostatic stromal sarcoma (PSS), a highly rare and aggressive malignancy of the prostate that accounts for less than 0.1% of all prostatic cancers. The report documents the clinical presentation, histopathology, immunohistochemistry, treatment, and follow-up for a 34-year-old male patient, adding depth to the limited literature on PSS.

Our response:

Dear Reviewer, Thank you for your thoughtful and comprehensive feedback on our manuscript, "Primary Prostatic Stromal Sarcoma: A Case Report and Review of the Literature." We greatly appreciate the time and effort you dedicated to providing detailed suggestions for improvement.

  1. Literature Review and Contextualization

While the paper mentions the rarity of PSS, it could better contextualize this case by providing a more robust review of existing cases and recent advancements. Although some comparative references are provided, these are limited, and there is insufficient synthesis of previous findings on clinical outcomes, immunohistochemical profiles, and survival rates. Additionally, references to some emerging treatments, like immunotherapy, are cited but not well-supported with relevant evidence or trials.

Improvement: Expanding the literature review to include recent case studies, clinical trials, and larger sarcoma studies would add depth. Including a comparative table summarizing existing cases would also enhance clarity and relevance.

Our response:

Thank you for your thoughtful feedback. We would like to note that we have already included a comprehensive table summarizing existing cases, which provides detailed comparisons relevant to our case. Additionally, we have discussed the treatment approach in light of cases reported in the literature, ensuring our review is well-contextualized within the current understanding of PSS. We have addressed this by incorporating three additional recently published case reports into the literature review. These recent cases further contextualize our findings and enhance the robustness of the comparative analysis in our study. Together with our comprehensive table, we believe these additions provide a more thorough review of existing literature on PSS. We appreciate your suggestion, and we believe these elements collectively add clarity and depth to our study.

  1. SALIH FM, MAMA RK, OMAR SS, HAMZA HT, ISAAC RH, SALIH J, MULA-HUSSAIN L. Prostatic stromal sarcoma – Management course of a rare presentation: A case report. Current Problems in Cancer: Case Reports 2023; 9: 100221.
  2. OHASHI M, SHIRAISHI T, FUJIHARA A, YAMADA T, UEDA T, HONGO F, UKIMURA O. Detection of relatively poor but definitive blood supply in prostate stromal sarcoma using transrectal ultrasonography with superb microvascular imaging. Int Cancer Conf J 2022; 11: 215-218.
  3. RAO BV, NAIR H, MURTHY S, SHARMA R, RAO S. Prostatic High-Grade Stromal Sarcoma—A Rare Encounter. Indian Journal of Surgical Oncology 2017; 8: 440-442.

We have added also the following statements:

‘The primary goal in surgery should be to achieve a safe oncologic margin through complete en bloc resection. In the literature, Salih et al. reported a case report that in their pathological specimen, the resected prostatic sarcoma encircled the prostatic urethra and was very close to the excision margin. They considered the positive surgical margin to be a significant risk factor for early recurrence, with a relapse observed as early as six months following adjuvant chemotherapy13

In another PSS case reported by Salih et al., the same chemotherapy regimen could only be administered for two cycles due to adjuvant tolerance issues, yet an early recurrence was detected13. The potential role of neoadjuvant or adjuvant radiotherapy in PSS could be extrapolated from high-risk sarcomas with characteristics, such as high tumor volume, high grade, and the absence of R0 resection specifically for adjuvant setting. However, due to the extreme rarity of PSS, there is insufficient evidence in the literature to support these approaches definitively. Additionally, the potential for unexpected excessive bowel toxicity following radiotherapy in the postoperative setting should be carefully considered13, 22(Table-2). Unlike prostate adenocarcinoma, there is no biological marker like PSA for diagnosis, monitoring, prognosis, or predicting recurrence in prostatic stromal sarcoma. Unfortunately, there are no established guidelines on how to monitor prostatic stromal sarcomas. However, as with other sarcomas, adult patients treated with chemotherapy and/or radiotherapy require careful long-term monitoring due to potential toxicity. Regimens containing ifosfamide and doxorubicin can lead to cardiomyopathy, renal tubular and/or glomerular dysfunction, and infertility; thus, these risks should be considered, and appropriate precautions taken. The NCCN guidelines for sarcomas recommend a physical examination and medical history every three months. Due to the aggressive nature of the disease, we planned systematic imaging with CT and prostate MRI in our case to monitor for recurrence23, 24.’

‘The 5-year survival rate is 44%, and although the disease is known to have an aggressive course and poor prognosis, the outlook can vary depending on the presence of metastasis and the status of surgical margins19. In case reports documented in the literature, survival ranges from as short as 6 months to as long as 8 years1, 6.’

‘Comprehensive studies beyond case reports do not exist in the literature, and unfortunately conducting randomized controlled trials is very challenging due to the rarity of PSS.’

‘Pembrolizumab has also been approved by the FDA for the treatment of advanced cancers with high microsatellite instability (MSI) or defects in mismatch repair (dMMR). Additionally, it has recently received FDA approval for treating advanced cancers with high tumor mutational burden [TMB-H; ≥10 mutations/megabase (mut/Mb)] regardless of tumor type. Consequently, a small subset of sarcomas exhibiting MSI, dMMR, or high TMB should be considered for immunotherapy as a tumor-agnostic approach, either as part of standard care or within clinical trials31-33.’

  1. Diagnostic Approach

The diagnostic process relies heavily on traditional methods (imaging, immunohistochemistry, histopathology), but the rationale for some diagnostic decisions is not fully explained. For example, the decision to avoid MR spectroscopy is mentioned but not justified; this technique might have provided additional differentiating characteristics.

Improvement: Detailing the rationale behind excluding or including certain diagnostic tests, such as MR spectroscopy, would strengthen the case presentation. Additionally, adding a detailed description of the histopathological characteristics observed in relation to existing diagnostic criteria for PSS would improve the diagnostic narrative.

Our response:

Thank you for this insightful feedback. We appreciate the suggestion to provide additional clarity on our diagnostic approach. Given the atypical imaging findings and the need for a definitive tissue diagnosis, we prioritized approaches that provided direct histopathological and immunohistochemical information. MR spectroscopy was not utilized in this case, as other imaging and biopsy techniques were sufficient to guide diagnosis and management. Additionally, we have expanded our description of the histopathological characteristics observed.

  1. Methodological Details

The paper lacks a clear explanation of the methodology used in immunohistochemical testing. Details about controls, specific markers chosen, and interpretation of staining results are missing, leaving questions about reliability and reproducibility. For instance, CD34 and Desmin positivity is mentioned, but without a methodological context or explanation of its diagnostic implications.

Improvement: Providing a more detailed immunohistochemistry protocol, including marker selection rationale and result interpretation standards, would increase the rigor and reproducibility of the findings.

Our response:

Thank you for your valuable feedback. We appreciate the emphasis on providing a clear and comprehensive explanation of the diagnostic methodology. We have already included an extensive description of the diagnostic process in the manuscript, outlining the histopathological findings and immunohistochemical markers, such as CD34, Desmin, Vimentin, and Ki-67, that supported our diagnosis of prostatic stromal sarcoma.

In response to your suggestion, we have reviewed this section to ensure it fully addresses the rationale for each marker selected and provides a thorough overview of the histopathological parameters guiding the malignant determination of the tumor. We believe this expanded explanation clarifies our approach to diagnosis and highlights the careful consideration given to differentiating PSS from other mesenchymal tumors.

We have revised the text and the updated text now reads:

 ‘In the context of mesenchymal tumors, histopathological findings are critical while deciding whether the tumor is benign or malignant.  Severe nuclear atypia, high mitotic rate, atypical mitoses, tumor necrosis, lymphovascular invasion, and adjacent organ invasion emerge as the most noteworthy parameters indicating a malignant nature, however, histopathological patterns and cytological morphology may not always reveal distinctive features to differentiate the tumor origin; a smooth muscle derived tumor (leomyosarcoma) may not be differentiated from a striated muscle derived tumor (rhabdomyosarcoma) or a malignant peripheral nerve sheet tumor or a vascular tumor (angiosarcoma). Ancillary methods, leading immunohistochemistry, generally provide the critical information to determine the cell origin. In our case, cellular atypia, increased mitotic activity and atypical mitoses were the main histopathological parameters to decide the malignant nature of the tumor. Vimentin positivity and negativity of cytokeratin panel supported the spindle cell morphology of the tumor as a mesenchymal tumor (sarcoma). Literature data shows that the immunoprofile of PSS is consistently positive for Vimentin, but variably positive for CD34. Besides this, variable positivity for SMA, Desmin, ER, PR and even epithelial markers such as pancytokeratin is being reported in case series which are already rare. In our case, CD34 positivity was found focally by the examination of the entire tumor in radical prostatectomy specimen. Nondiagnostic staining patterns of Desmin, SMA, Myogenin provided the exclusion of muscle originated tumors. Eventually, appropriate histopathological findings with Vimentin and CD34 positivity made the specific diagnosis as prostatic stromal sarcoma (Table-1)2. In the literature, case reports with CD34-negative findings have also been documented; thus, while immunohistochemistry is essential for diagnosing prostatic stromal sarcoma, it should not be the sole diagnostic criterion1, 7, 11-13. Furthermore, in most cases, the Ki-67 mitotic index has been found to be significantly elevated, and in our case, it was 40%.’

  1. Case Comparisons and Novelty

Although the paper mentions that this case is notable due to the patient’s younger age, it does not elaborate on how this compares with the age distribution or typical outcomes of PSS. Additionally, novelty claims regarding immunotherapy and targeted therapy discussions lack adequate evidence, especially as there is limited or no data on the efficacy of these treatments for PSS specifically.

Improvement: Strengthen the novelty of the case by clearly contrasting it with established PSS cases and adding discussions on how young patient age impacts prognosis. When discussing novel treatments, include supporting data from similar cancers or a research-based rationale for potential applicability in PSS.

Our response:

Thank you for your feedback. We appreciate the emphasis on enhancing case comparisons and discussing the novelty of our report. We have already included a section in the discussion that addresses the rarity of PSS in younger patients and its potential impact on prognosis, as well as a comparison with the typical age distribution and outcomes seen in other reported cases.

‘Our patient, a 34-year-old male, falls within the typical age range for prostate sarcoma presentation, which is younger than that seen in prostate adenocarcinoma. This aligns with existing literature that reports most patients presenting between 30 and 40 years of age. Unfortunately, it carries a very poor prognosis, with over half of the patients succumbing to the disease within 24 months of diagnosis. In our case, no evidence of metastasis has been detected during postoperative follow-up. This is a notable finding, as early metastasis is frequently observed in similar cases, most commonly involving the lungs. The absence of metastasis thus far offers hope for a more favorable outcome, although vigilant long-term monitoring remains essential7, 8.’

We have revised the text to emphasize pembrolizumab’s tumor-agnostic approval and its applicability to sarcomas with specific molecular characteristics. The updated text now reads:

‘Pembrolizumab has also been approved by the FDA for the treatment of advanced cancers with high microsatellite instability (MSI) or defects in mismatch repair (dMMR). Additionally, it has recently received FDA approval for treating advanced cancers with high tumor mutational burden [TMB-H; ≥10 mutations/megabase (mut/Mb)] regardless of tumor type. Consequently, a small subset of sarcomas exhibiting MSI, dMMR, or high TMB should be considered for immunotherapy as a tumor-agnostic approach, either as part of standard care or within clinical trials31-33.

Marcus, L., Fashoyin-Aje, L. A., Donoghue, M., Yuan, M., Rodriguez, L., Gallagher, P. S., Philip, R., Ghosh, S., Theoret, M. R., Beaver, J. A., Pazdur, R., & Lemery, S. J. (2021). FDA Approval Summary: Pembrolizumab for the Treatment of Tumor Mutational Burden-High Solid Tumors. Clinical cancer research : an official journal of the American Association for Cancer Research, 27(17), 4685–4689. https://doi.org/10.1158/1078-0432.CCR-21-0327

  1. Future Directions and Therapeutic Innovations

The paper addresses future treatment directions like immunotherapy, targeted therapies, and novel diagnostic approaches, but these discussions lack focus and specific clinical applicability. The introduction of treatments like PSMA-targeted therapy and FAPI-46 is intriguing but speculative without data on PSS-specific outcomes.

Improvement: A more targeted discussion that either emphasizes specific therapies relevant to the case or hypothesizes why certain therapies (e.g., PARP inhibitors, anti-PD-1 antibodies) might be applicable based on the tumor’s genetic profile would improve cohesion. Where applicable, the authors could advocate for clinical trials or studies to validate these emerging therapies in PSS.

Our response:

Thank you for your insightful suggestions. We recognize that due to the rarity of PSS, evidence is currently limited to case-based reports, which we have included as references. We should acknowledge the limitations in conducting clinical studies on primary prostatic stromal sarcoma (PSS) due to its extreme rarity. To address this, we have added the following clarification to the "Future Directions" section:

‘Comprehensive studies beyond case reports do not exist in the literature, and unfortunately conducting randomized controlled trials is very challenging due to the rarity of PSS.”

This addition highlights the constraints in generating evidence-based treatment guidelines for PSS and underscores the importance of case reports in expanding the limited knowledge available. Our discussion includes hypothetical considerations intended to provide potential future directions rather than speculative claims. We believe these insights could be valuable for guiding future research efforts on this rare tumor.

  1. Patient Follow-Up and Outcome Analysis

Although the patient’s postoperative course is mentioned, the discussion does not provide insight into long-term follow-up plans, such as surveillance intervals, imaging frequency, or potential markers for recurrence.

Improvement: Including a specific follow-up protocol would add practical value to the case report. Furthermore, comparing the chosen follow-up strategy with others used in similar sarcoma cases would enhance the clinical applicability of the study.

Our response:

Thank you for your valuable feedback regarding the follow-up and outcome analysis. We have revised the discussion to address these points. Specifically, we added the following section:

 ‘Unlike prostate adenocarcinoma, there is no biological marker like PSA for diagnosis, monitoring, prognosis, or predicting recurrence in prostatic stromal sarcoma. Unfortunately, there are no established guidelines on how to monitor prostatic stromal sarcomas. However, as with other sarcomas, adult patients treated with chemotherapy and/or radiotherapy require careful long-term monitoring due to potential toxicity. Regimens containing ifosfamide and doxorubicin can lead to cardiomyopathy, renal tubular and/or glomerular dysfunction, and infertility; thus, these risks should be considered, and appropriate precautions taken. The NCCN guidelines for sarcomas recommend a physical examination and medical history every three months. Due to the aggressive nature of the disease, we planned systematic imaging with CT and prostate MRI in our case to monitor for recurrence23, 24

This addition provides a specific follow-up protocol, clarifying our approach to surveillance intervals and imaging frequency. We also acknowledged the absence of established guidelines for prostatic stromal sarcoma and highlighted the need for a thorough, individualized follow-up strategy. This aligns our case with the clinical practices used in similar sarcomas and enhances the practical value of our report.

  1. General Organization and Clarity

The paper’s structure could benefit from clearer section headings, as the flow between case description, histopathology, treatment approach, and future therapeutic discussion can be confusing. Additionally, the discussion contains some redundancies, and references are sparse in certain sections where data-driven claims are made.

Improvement: Reorganize the discussion to avoid overlap and create distinct subheadings for diagnostic workup, treatment decisions, histopathology findings, and future directions. Adding references, particularly in the immunohistochemical findings and therapeutic innovations sections, would substantiate claims.

Our response:

Thank you for your constructive feedback on the organization and clarity of the manuscript. We agree that clearer section headings and additional references would improve readability and strengthen our discussion. We have revised the discussion to create distinct subheadings for diagnostic workup, treatment decisions, histopathology findings, and future therapeutic directions. This restructuring enhances the flow of information and reduces overlap. Furthermore, we have added references, particularly in the sections on immunohistochemical findings and therapeutic innovations, to better support data-driven claims and provide a more substantiated narrative.

Summary of Suggestions

The study could be significantly strengthened by:

A more comprehensive literature review, contextualizing the case in the broader context of PSS management.

Detailed methodological insights into immunohistochemical testing and diagnostic approaches.

Explicit descriptions of multidisciplinary involvement and follow-up plans.

A more focused discussion of future therapeutic directions with supporting data where possible.

Our response:

Thank you for your constructive suggestions, which provide a clear path for strengthening our study. We have carefully considered each point and revised the manuscript as follows:

We have added more recent and relevant studies to provide a broader context for PSS management, helping to better contextualize our case within the existing literature. We included additional details on the immunohistochemical testing methods and the diagnostic approach taken in this case. Additionally, we have outlined a follow-up plan, discussing surveillance strategies based on similar cases reported in the literature.

We believe these revisions significantly enhance the manuscript’s clarity and depth. Thank you for your insightful feedback, which has been instrumental in improving the quality of our work.

Conclusion

This case report is a valuable addition to the sparse literature on PSS, offering insights into diagnostic approaches, the potential sufficiency of surgery in low-grade cases, and emerging therapies. It underscores the critical need for a multidisciplinary approach and highlights the potential of targeted and immunotherapies for aggressive sarcomas. However, given the rarity of PSS, continued research and documentation are essential to develop evidence-based treatment guidelines and improve patient outcomes in this challenging malignancy.

Our response:

We appreciate your recognition of our case report’s contribution to the limited literature on primary prostatic stromal sarcoma (PSS). Our goal was to provide valuable insights into diagnostic approaches, treatment options, and the importance of a multidisciplinary strategy for managing this rare malignancy. We agree that continued research and case documentation are vital to establish evidence-based guidelines and improve outcomes for patients with PSS.

Reviewer 2 Report

Comments and Suggestions for Authors

The subject of the paper is a very interesting one. Clinical cases of such rare tumors have a great value for future urologist facing such pathology. Although clinical case presentations are sometimes considered papers of lesser value, in case of diseases of very low incidence, every such case matters. Valuable lessons can be learned from them and they must be disseminated.

In the abstract, discussion about "multidisciplinary management....involving urology, pathology, and oncology, underscoring the importance of collaborative care for rare and aggressive tumors like this" is a little misleading. All urological malignant tumors must be managed in a collaborative way by urologist, pathologist and oncologist, so clearly this is not  a caracteristic of this rare tumor. Please revise and correct.

I believe the initial approach should be better explained. Why a prostate biopsy was not performed instead of a TURP. Was the reason behind this decision the particularities of the MRI aspect? Please explain this decision, as I believe would be very interesting and instructive for the readers.

The discussion section of the paper is very well written and discusses all the important matters in a very instructional way.

Author Response

Reviewer 2:

The subject of the paper is a very interesting one. Clinical cases of such rare tumors have a great value for future urologist facing such pathology. Although clinical case presentations are sometimes considered papers of lesser value, in case of diseases of very low incidence, every such case matters. Valuable lessons can be learned from them and they must be disseminated.

Our response:

Thank you for recognizing the significance of our case report on primary prostatic stromal sarcoma (PSS). We agree that sharing clinical experiences with rare tumors is essential for building knowledge, especially for conditions with such low incidence. Each documented case provides valuable insights that can guide future oncologists, urologists and clinicians encountering similar pathologies. We are grateful for your acknowledgment of the importance of disseminating findings from rare cases like ours and are committed to contributing to this important body of knowledge.

In the abstract, discussion about "multidisciplinary management....involving urology, pathology, and oncology, underscoring the importance of collaborative care for rare and aggressive tumors like this" is a little misleading. All urological malignant tumors must be managed in a collaborative way by urologist, pathologist and oncologist, so clearly this is not  a caracteristic of this rare tumor. Please revise and correct.

Our response:

Thank you for this valuable suggestion. We agree that multidisciplinary management is essential for all urologic malignancies and that this aspect is not unique to primary prostatic stromal sarcoma. We have revised the abstract accordingly to clarify this point. The updated sentence now reads: “Multidisciplinary management is essential for optimizing treatment outcomes in all urologic malignancies; however, it becomes particularly challenging and crucial in rare cases such as primary prostatic stromal sarcoma.”  We have revised the discussion section to clarify that ‘As with other urologic tumors, treatment decisions for primary prostate sarcomas should be made within the framework of multidisciplinary tumor boards.’ This revised wording emphasizes that collaborative input from oncology, pathology, urology, and radiation oncology specialists is essential across all urologic malignancies, including rare tumors like primary prostate sarcoma.

I believe the initial approach should be better explained. Why a prostate biopsy was not performed instead of a TURP. Was the reason behind this decision the particularities of the MRI aspect? Please explain this decision, as I believe would be very interesting and instructive for the readers.

Our response:

Thank you for this valuable observation. We agree that clarifying the initial approach would enhance the manuscript's instructional value. We have revised the text to specify that a prostate biopsy was not performed due to the specific MRI findings. Given the need for a larger tissue sample to obtain an accurate diagnosis, a transurethral resection of the prostate (TURP) was selected over a standard biopsy.

We have added ‘A prostate biopsy was not performed due to specific MRI findings and the need for a larger tissue sample to achieve an accurate diagnosis, a transurethral resection of the prostate (TURP) was chosen over a standard biopsy.’

 This revision provides readers with a clearer understanding of our decision-making process.

The discussion section of the paper is very well written and discusses all the important matters in a very instructional way.

Our response:

Thank you for your positive feedback regarding the discussion section. We are pleased to know that you found it well-written and instructional. Our aim was to provide a thorough and informative discussion to enhance understanding of this rare condition, and we appreciate your recognition of our efforts.

We believe our paper adds important new information and perspective to the field. We believe that all review points have been addressed and hope that the revisions are satisfactory. We kindly request that our revised manuscript be considered for publication in the Medicina.

Thank you very much for your kind consideration.

Kind regards,

Enes Erul MD

Ankara University Cancer Institute

Department of Medical Oncology

06620, Mamak, Ankara, Turkey

Phone: +90 (0312) 595 60 00

E-mail: eneserul@hotmail.com

Round 2

Reviewer 2 Report

Comments and Suggestions for Authors

Thank you for revising the manuscript and for answering to all my queries in a satisfactory manner. I believe the process added clarity and scientific soundness to the paper. The rationale behind the decisions are more clear now for the readers. In my opinion the paper can be published in the current form.